# Laryngeal Cancer Cells Metabolize 25-Hydroxyvitamin D_3_ and Respond to 24R,25-dihydroxyvitamin D_3_ via a Mechanism Dependent on Estrogen Receptor Levels

**DOI:** 10.3390/cancers16091635

**Published:** 2024-04-24

**Authors:** Cydney D. Dennis, Jonathan T. Dillon, Prit H. Patel, David J. Cohen, Matthew S. Halquist, Adam C. Pearcy, Barbara D. Boyan, Zvi Schwartz

**Affiliations:** 1Department of Biomedical Engineering, Virginia Commonwealth University, Richmond, VA 23284, USA; denniscd@vcu.edu (C.D.D.); dillonj4@vcu.edu (J.T.D.); patelph9@vcu.edu (P.H.P.); djcohen@vcu.edu (D.J.C.); bboyan@vcu.edu (B.D.B.); 2Department of Pharmaceutics, Virginia Commonwealth University, Richmond, VA 23298, USA; halquistms@vcu.edu (M.S.H.); adamcpearcy@gmail.com (A.C.P.); 3Bioanalytical Core Laboratory, Central Virginia Drug Abuse Research Center, Virginia Commonwealth University, Richmond, VA 23298, USA; 4Wallace H. Coulter Department of Biomedical Engineering, Georgia Institute of Technology, Atlanta, GA 30332, USA; 5Department of Periodontics, The University of Texas Health Science Center at San Antonio, San Antonio, TX 78229, USA

**Keywords:** vitamin D metabolism, head and neck cancer, estrogen receptor, local production

## Abstract

**Simple Summary:**

Vitamin D supplementation’s effectiveness in reducing cancer is widely debated. The controversy may be explained by differences in 25(OH)D_3_ metabolism. Previous studies showed that breast cancer cells produce vitamin D_3_ metabolites locally, and their ability to regulate tumorigenesis is related to the presence of estrogen receptor alpha 66 (ERα66). The present study examined this process in another estrogen-dependent cancer, laryngeal cancer. This study evaluated the effects of the active vitamin D_3_ metabolite 24R,25(OH)_2_D_3_ on cell growth, whether these cells can produce their own secreted vitamin D_3_ metabolites, and whether this ability is related to the presence or absence of ERα66. To address these questions, ERα-positive (UM-SCC-12) and ERα-negative (UM-SCC-11A) laryngeal cancer cell lines were examined for the presence of ERα66; the enzymes responsible for metabolizing 25(OH)D_3_ (CYP24A1 and CYP27B1); the local production of 24,25(OH)_2_D_3_ and 1,25(OH)_2_D_3_; and the effect of 24R,25(OH)_2_D_3_ on cell growth and markers of metastasis. The results showed that laryngeal cancer cells express CYP24A1 and CYP27B1, produce 24,25(OH)_2_D_3_, and respond differently to 24R,25(OH)_2_D_3_, in correlation with ERα66 levels. These findings suggest that tumor cells locally produce vitamin D metabolites to regulate their growth, which must be considered when recommending vitamin D supplementation.

**Abstract:**

Studies have evaluated vitamin D_3_’s therapeutic potential in estrogen-responsive cancers, with conflicting findings. We have shown that the proliferation of breast cancer cells is regulated by 24R,25-dihydroxyvitamin D_3_ (24R,25(OH)_2_D_3_) depending on estrogen receptor alpha 66 (ERα66) expression, suggesting that this could also be the case for estrogen-sensitive laryngeal cancer cells. Accordingly, we examined levels of ERα isoforms in ERα66-positive UM-SCC-12 and ERα66-negative UM-SCC-11A cells and their response to 24R,25(OH)_2_D_3_. 24R,25(OH)_2_D_3_ stimulated proliferation, increased the expression of metastatic markers, and inhibited apoptosis in UM-SCC-12 cells while having the opposite effect in UM-SCC-11A cells. To evaluate if vitamin metabolites could act via autocrine/paracrine mechanisms, we assessed the expression, protein levels, and activity of vitamin D_3_ hydroxylases CYP24A1 and CYP27B1. Both cell types expressed both mRNAs; but the levels of the enzymes and their activities were differentially regulated by estrogen. ERα66-negative UM-SCC-11A cells produced more 24,25(OH)_2_D_3_ than UM-SCC-12 cells, but comparable levels of 1,25(OH)_2_D_3_ when treated with 25(OH)D_3_ These results suggest that the regulation of vitamin D_3_ metabolism in laryngeal cancer cells is modulated by ERα66 expression, and support a role for 24R,25(OH)_2_D_3_ as an autocrine/paracrine regulator of laryngeal cancer. The local metabolism of 25(OH)D_3_ should be considered when determining the potential of vitamin D_3_ in laryngeal cancer.

## 1. Introduction

Laryngeal cancer accounts for approximately one-third of all head and neck cancers. The American Cancer Society estimates men to be four times more likely to be diagnosed with laryngeal cancer and die from it compared to women in the United States [1]. Approximately 60% of patients are diagnosed at advanced stages (stage III or IV), at which point surgery with adjuvant chemotherapy or radiotherapy is predominantly used [1,2]. The constant use of aggressive treatments stresses the need for further research into alternative therapies and early-stage biomarkers.

Previous epidemiological studies, along with the significant difference in sex susceptibility in laryngeal cancer, have raised questions about the role of sex hormones in malignancy. At the same time, many studies have shown that smoking and alcoholism are some of the most common contributing factors to laryngeal cancer [3]. However, global incidence was demonstrated to be seven times higher in men compared to women regardless of alcohol or tobacco usage [4,5]. The evident gender dispositions found in the larynx, physiological changes observed during puberty, and the organ characterized as a secondary sex organ imply that sex hormones or endocrine factors could play a significant role in the tumorigenesis of laryngeal cancer [3,6]. However, it is still widely debated as a sex hormone-dependent cancer.

Researchers observed the potential role of estrogen in laryngeal cancer when examining the change in estrogen levels in men over the age of 60, correlating to the age at which men have an increased risk of incidence [7]. Other studies demonstrated a higher probability of abnormal estrogen metabolism in a cohort with 60% laryngeal cancer patients compared to healthy patients [8]. The presence of functional estrogen receptors in laryngeal cancer and its estrogen dependence have been contested for years [9,10,11]. They were first described in laryngeal cancer over 30 years ago, when studies showed an increase in the reactivity of sex hormone receptors, specifically ERα, in malignant laryngeal tissue compared to adjacent healthy tissue [6,12,13], and many studies have shown that laryngeal cancers respond to estrogen [14,15,16]. The controversy involving laryngeal cancer as an estrogen-responsive cancer could be due to the presence or absence of ERα66 and/or its splice variants ERα36 and ERα46.

The role of estrogen in tumorigenesis is well established in hormone-responsive cancers like breast and prostate cancers [17]. Estrogen regulates proliferation, survivability, and metastasis via its receptor, estrogen receptor alpha-66 (ERα66). As a result, ERα66 is currently used as a critical molecular biomarker in diagnostics, predicting prognoses and influencing treatment [1,16,18].

ERα66 has been thoroughly investigated in breast cancer, demonstrating its role in mediating the effects of estrogen and specific hormone therapies on tumorigenesis [17,19]. Most commonly, clinicians use the presence of ERα66 in breast cancer as a well-established prognostic factor for aggression and responsiveness to endocrine therapy [9,20]. It is also a common target for hormone therapy treatments like tamoxifen [14]. On the other hand, ERα66-negative breast cancers are more aggressive and tend to be more resistant to hormone therapy treatment and, therefore, require other treatment modalities.

Recently, studies have investigated the role of ERα66 splice variants in mediating breast cancer response to estrogen [15,17,19]. ERα36 contains ligand binding domains similar to ERα66 but does not include the transcriptionally active binding domains AF-1 and AF-2 [19,21,22]. It is located in the plasma membrane and is expressed in both ERα66-positive and ERα66-negative breast cancers. Estrogen binding to ERα36 induces phospholipase C (PLC)-dependent protein kinase C (PKC) signaling through a caveolae membrane-mediated mechanism to enhance metastasis and regulate tamoxifen resistance [21,22,23]. ERα46 is a lesser studied isoform due to its similarity in structure to ERα66, but it does not include the AF-1 transactivation domain [19,22,24]. Studies suggest that its similar structure contributes to its function in that it may act as a competitive inhibitor of ERα66, but more studies are needed to confirm this hypothesis [22,24].

Similar to breast cancers [15,17,19], the presence of ERα66 in laryngeal cancers can determine the tumor’s aggressiveness and how the cells will respond to treatments with estrogen [16,25]. In a previous study, ERα66-positive UM-SCC-12 cells and ERα66-negative UM-SCC-11A cells, which came from male donors, exhibited differential responsiveness to 17β-estradiol; the resulting changes in tumor aggressiveness were dependent on the expression of ERα66 [16,25]. In human laryngeal tissue samples, high levels of ERα66 were measured in samples of earlier stage tumors, whereas stage IV tumor samples exhibited the lowest ERα66 levels. There was no correlation between staging and ERα36 expression, even though the receptor was shown to play a role in estrogen-mediated tumorigenesis and chemoresistance [15,25]. These data suggest that ERα66 plays a critical role in tumor aggression, particularly in early-stage cancers.

In vitro studies using ER-positive UM-SCC-12 cells and ER-negative UM-SCC-11A cells support this hypothesis [16,25]. 17β-estradiol increased proliferation in UM-SCC-12 cells but reduced proliferation in UM-SCC-11A cells. Treatment inhibited apoptosis in UM-SCC-12 cells, but there was no change in apoptosis in UM-SCC-11A cells. These data demonstrate the variability in response to estrogen in laryngeal cancer cells and suggest that this may be due at least in part to the difference in ERα66 expression [16]. Therefore, ERα66 may be a promising prognostic and diagnostic biomarker in laryngeal cancer similar to the findings in breast cancer.

The seco-steroid vitamin D_3_ also impacts tumor growth and metastasis. Vitamin D_3_ deficiency is associated with an increased risk of malignancy in multiple cancers [26,27,28]. In addition, vitamin D_3_ supplementation (25(OH)D_3_) is used as a therapeutic agent in various cancers, like breast, prostate, colorectal, and ovarian cancer. Evidence shows that vitamin D_3_ can affect tumorigenesis by regulating the innate and adaptive immune systems [28]. Vitamin D deficiency could explain the racial disparities in laryngeal cancer, suggesting that vitamin D_3_ could also be a risk factor for early-age diagnosis. African American men are at a higher risk of diagnosis at a younger age and death than Caucasian men [2]. Randomized trials demonstrated that despite similar smoking rates compared to Caucasian men, a common risk factor for laryngeal cancer, African American men were at the highest risk of vitamin D_3_ deficiency [29].

These studies suggest that vitamin D_3_ supplementation may be a promising therapeutic agent in laryngeal cancer treatment. Whereas many reports demonstrate that vitamin D deficiency is not a risk factor for laryngeal cancer [30,31], other reports show that high levels of circulating 25(OH)D_3_ are associated with a lower incidence of malignancy [32]. Several factors may contribute to this difference, including hormone receptor expression, seasonal measurements, downstream vitamin D_3_ metabolism, and localized metabolite production in the tissue [31]. These factors need to be assessed to determine the therapeutic potential of vitamin D_3_ in laryngeal cancer.

Previous studies in breast cancer showed that the vitamin D_3_ metabolite, 24R,25(OH)_2_D_3_, can regulate tumorigenesis without destabilizing serum calcium levels, unlike 1α,25(OH)_2_D_3_. The effect of 24R,25(OH)_2_D_3_ on tumorigenesis depended on the expression of ERα66. 24R,25(OH)_2_D_3_ inhibited tumorigenesis in ERα66-positive MCF7 cells but increased tumorigenesis in the ERα66-negative HCC38 cell line [33]. We examined the potential for autocrine-based effects causing the difference in response by evaluating the expression of hydroxylase enzymes and the production of vitamin D_3_ metabolites. The expression of CYP27B1, the hydroxylase that is responsible for converting 25(OH)D_3_ into 1,25(OH)_2_D_3_, in healthy tissue indicated that breast cancer cells could locally produce metabolites of vitamin D_3_, suggesting that vitamin D_3_ signaling could be regulated in an autocrine or paracrine manner [33,34,35]. However, it was unclear how the 24-hydroxylase, CYP24A1, the enzyme responsible for converting 25(OH)D_3_ into 24,25(OH)_2_D_3_, was expressed or how the expression of both enzymes was related to ERα expression.

To elucidate the conflicting findings between serum vitamin D levels and laryngeal cancer progression, the present study aimed to evaluate the effect of the active vitamin D3 metabolite, 24R,25(OH)_2_D_3_, on markers of laryngeal tumorigenesis, determine if the expression of ERα66 is correlated with this effect in two different laryngeal cancer cell lines, and ascertain if these cells can regulate their tumorigenesis in an autocrine manner. We first examined the protein levels of ERα isoforms in both cell types to confirm a difference in ERα66 mRNA levels. We then treated the ERα66-positive and ERα66-negative cells with 24R,25(OH)_2_D_3_ to determine their response. We assessed the possibility of an autocrine effect by examining the presence of hydroxylase enzymes responsible for the hydroxylation of 25(OH)D_3_ and whether their levels are affected by estrogen. Finally, we examined the production of two vitamin D_3_ metabolites, 24,25(OH)_2_D_3_ and 1,25(OH)_2_D_3_.

## 2. Materials and Methods

### 2.1. Cell Culture

Human male laryngeal squamous-cell carcinoma cell lines, UM-SCC12 (RRID: CVCL7717) and UM-SCC-11A (RRID: CVCL_7715), were developed in the Carey Laboratory of the Department of Otolaryngology/Head and Neck Surgery at the University of Michigan and purchased from the university. Six weeks prior to receiving them, the cells were sent to the American Type Culture Collection (ATCC, Manassas, VA, USA) for authentication via short tandem repeat testing (STR) and mycoplasma detection. Once received, the cells were expanded and passaged at least 3 times before being used for experiments. Cells were cultured in high-glucose Dulbecco’s Modified Eagle Medium (DMEM; GIBCO™, Fisher Scientific, Hampton, NH, USA) lacking phenol red and supplemented with 10% heat-inactivated fetal bovine serum (FBS), 2mM L-glutamine, 1% non-essential amino acids, and 100 U/mL penicillin-streptomycin (Thermo Fisher Scientific, Waltham, MA, USA). Male rat kidney epithelial cells (NRK-52E; RRID: CVCL_0468) were purchased from ATCC (CRL-1571) and cultured in high-glucose DMEM supplemented with 10% heat-inactivated FBS and 1% penicillin/streptomycin. Once received, the cells were expanded and passaged 3 times prior to use for experimentation.

### 2.2. Characterization of Estrogen Receptors

Cells were seeded in a 6-well plate at 10,000 cells/cm^2^ (n = 3 wells/cell type) and grown until fully confluent. At confluence, media were replaced one last time. After 24 h, the cells were washed and lysed in a 200 μL/well of radioimmunoprecipitation assay (RIPA) buffer. The cell lysate proteins were resolved on 4–20% Tris-glycine extended gels by means of gel electrophoresis (Bio-Rad, Hercules, CA, USA) and transferred onto a PVDF membrane using the Bio-Rad Mini Trans-blot^®^ Electrophoretic Transfer Cell. The membranes were then blocked and probed with a polyclonal ERα antibody (Thermo Fisher Scientific, RRID: AB_325814, CAT #PA1-309) along with a monoclonal GAPDH antibody (Millipore-Sigma, St. Louis, MO, USA, CAT #MAB374, Clone #6C5). Li-Cor IRDye^®^ Goat anti-rabbit 800 CW (CAT# 926-32211, RRID: AB_621843) and Goat anti-mouse 680RD (CAT# 926-68070, RRID AB_10956588) secondary antibodies (Li-COR Biosciences, Lincoln, NE, USA) were used to visualize the proteins. The membranes were imaged using the LiCor-Biosciences Odyssey^®^ CLx Infrared Imaging System.

The mRNA expression of estrogen-associated receptors and the vitamin D receptor were also measured as previously described [31]. UM-SCC-12 and UM-SCC-11A cells were plated on 24-well plates (n = 6) and cultured until confluence. At confluence, media were changed one last time, and then the cells were harvested 12 h later. RNA was extracted using Trizol (Invitrogen, Waltham, MA, USA), and quantified using the Take3 Microvolume Plate (Biotek, Winooski, VT, USA). The RNA was then used to synthesize cDNA libraries (Life Technologies, Carlsbad, CA, USA, CAT#4368813). The levels of ERα66 were measured using a primer that amplifies the first 173 amino acids of exon 1 in ESR1 [36]. Due to the structure of ERα46, it is not possible to distinguish ERα46 from ERα66; therefore, another primer was used, designed to amplify exons 7 and 8, found in both ERα46 and ERα66 but not in ERα36 [36]. A primer that amplified the 9th exon of ESR1 measured the levels of ERα36. The expression levels of ESR2 (ERβ), GPR30, and the classical vitamin D receptor (VDR) were also measured. Levels were quantified by means of real-time quantitative PCR using Power SybrGreen^®^ Master Mix (Applied Biosystems, Walthalm, MA, USA) and its associated primers designed through Thermo Fisher Scientific (Appendix A). The expression of all receptors was normalized to GAPDH expression.

### 2.3. Response to 24R,25(OH)_2_D_3_

24R,25(OH)_2_D_3_ was purchased from Enzo Life Sciences (Farmingdale, NY, USA, BML-DM300) to assess the effect of 24R,25(OH)_2_D_3_ on markers of tumorigenicity. UM-SCC-12 and UM-SCC-11A cell monolayers were first cultured on 96-well or 24-well plates (n = 6 wells/group). At the time of treatment, the cells were treated with 10^−7^ M, 10^−8^ M, 10^−9^ M 24R,25(OH)_2_D_3_, or vehicle for 15 min, at which point the media were replaced with fresh full media for 24 h. The cells were then assessed for proliferation via the EdU DNA synthesis assay (Thermo Fisher Scientific), and 3-(4,5-dimethylthiazol-2-yl)-2,5-diphenyltetrazolium bromide, (MTT) (Sigma-Aldrich, St. Louis, MO, USA). Apoptosis was measured via the p53 assay (R&D Systems, Minneapolis, MN, USA), DNA fragmentation (TUNEL assay) (Trevigen, Gaithersburg, MD, USA), BAX/BCL2 mRNA, and Caspase 9 Activity (Promega, Madison, WI, USA). Metastasis was measured through the Western blot analysis of Nidogen-1 and the ratio of osteoprotegerin to RANKL, two lung and bone metastasis markers, respectively.

#### 2.3.1. Cell Proliferation

##### DNA Synthesis

UM-SCC-12 and UM-SCC-11A cells were cultured in 96-well plates until 70% confluence and then serum-starved in full media lacking FBS for 48 h. After 48 h of serum starvation, the cells were treated as described above and incubated in fresh media for 20 h. At that point, the cells were pulsed with 10 μL/well of 1:100 diluted 5-ethynyl-2′-deoxyuridine (EdU) and then incubated for another 4 h. Finally, the cells were harvested and assessed for proliferation according to the manufacturer’s protocol (Thermo Fisher Scientific).

##### 3-(4,5-Dimethylthiazol-2-yl)-2,5-diphenyltetrazolium bromide-MTT Assay

In a separate experiment assessing the effects of 24R,25(OH)_2_D_3_ on cell viability, UM-SCC-12 and UM-SCC-11A cells were cultured in 96-well plates until confluence and treated as stated previously. After 24 h, the cells were harvested and examined for formazan crystal solubilization according to the MTT assay protocol (Sigma-Aldrich).

#### 2.3.2. Apoptosis

##### TUNEL Assay

Cells were cultured in 96-well plates (n = 6) until confluence and treated as described above. Twenty-four hours after treatment, DNA fragmentation was measured via colorimetric terminal deoxynucleotidyl transferase (TdT)-mediated dUTP nick end labeling (Trevigen, TiterTAC™ in situ microplate TUNEL assay) assay according to the manufacturer’s protocol.

##### BAX/BCL2 Gene Expression

The ratio of apoptotic markers B cell lymphoma protein-2-associated X protein (BAX) and B cell lymphoma protein 2 (BCL2) demonstrates the life or death response of cells to an apoptotic stimulus [31]. An increase in the BAX/BCL2 ratio indicates a decrease in resistance to apoptotic stimuli, leading to an increase in cell death [37]. Cells were harvested 12 h after treatment and assessed for gene expression as described above. BAX and BCL2 mRNA expression were first normalized to GAPDH and then the ratio of BAX to BCL2 was calculated. 

##### Total p53 Content

Total p53 content was measured to evaluate apoptosis. UM-SCC-12 and UM-SCC-11A cells were plated in 24-well plates and treated as described above. Twenty-four hours after treatment, the cells were lysed in 500 μL 1mM EDTA, 0.05% Triton™X-100. The total p53 content was measured in the cell lysate and normalized to total protein content (Pierce 660nM BCA Protein Assay, Thermo Fisher Scientific) according to the manufacturer’s protocol.

##### Caspase 9 Activity

The initiator caspase, caspase-9, plays a critical role in the intrinsic pathway, activating effector caspases, which initiate the degradation phase of apoptosis [38]. Caspase 9 activity was measured 24 h after treatment using the Caspase-Glo^®^ 9 Assay (Promega, CAT #G8210) according to the manufacturer’s protocol. 

##### Markers of Metastasis

Two indicators of distant metastasis were measured to examine the effects of 24R,25(OH)_2_D_3_ on markers of metastasis. 

##### Nidogen-1 Lung Metastasis

The lung is the most common site of metastasis in laryngeal cancer [39]. To evaluate the effects of 24R,25(OH)_2_D_3_ on metastasis, we used a common marker of lung metastasis, nidogen-1. Nidogen-1 is a glycoprotein in the basement membrane that plays a role in the cross-linking of collagen IV and laminin [40]. Studies demonstrated that nidogen-1 activates integrin signaling in endothelial cells to thereby induce the transmigration of cancer cells to the lung parenchyma [41]. Nidogen-1 was measured by means of Western blot analysis 24 h after cells were treated as described above. Membranes were probed with a monoclonal anti-nidogen-1 primary antibody (Abcam, Cambridge, UK, CAT#ab133686). Nidogen-1 protein levels were normalized to the intensity of GAPDH. 

##### Osteoprotegerin/RANKL Bone Metastasis

Bone is another common site of metastasis in laryngeal cancer, which causes osteolytic lesions leading to fragmentation [39]. Osteoprotegerin (OPG) is a decoy receptor for the receptor activator of nuclear factor kappa B (RANK) ligand (RANKL); when OPG binds RANKL, it prevents the binding of the ligand to RANK on the surface of pre-osteoclasts, thereby prevent their terminal differentiation to bone-resorbing osteoclasts. The ratio of OPG to RANKL provides an indicator of bone resorptive potential; a higher osteoprotegerin/RANKL suggests reduced resorption. Confluent cultures of UM-SCC-12 and UM-SCC-11A cells were treated as previously described. At 24 h post treatment, conditioned media were collected and assayed for OPG (R&D Systems, Minneapolis, MN, CAT #DY805) and RANKL (PeproTech, Cranbury, NJ, USA, CAT #900-K142). Total DNA content was measured in the cell layer lysates lysed in 0.5% Triton™X-100. Protein levels of OPG and RANKL were first normalized to the total DNA content (Promega QuantiFluor* dsDNA, VWR, Radnor, PA, USA, CAT# PAE2670) and then calculated as the OPG/RANKL ratio. 

##### 24-Hydroxylase and 1-Hydroxylase Measurements

Quantitative PCR was used to examine the levels of CYP24A1 and CYP27B1 in both ERα66-positive UM-SCC-12 and ERα66-negative UM-SCC-11A cells. 25(OH)D_3_ is hydroxylated to 1,25(OH)_2_D_3_ or 24,25(OH)_2_D_3_ in the kidneys; therefore, NRK-52E rat kidney epithelial cells were used as a positive control for baseline levels of the 24-hydroxylase CYP24A1 and the 1-hydroxylase CYP27B1 enzymes as described above [31]. CYP24A1 and CYP27B1 expression levels in confluent cultures were normalized to the expression levels of GAPDH. 

To measure the amounts of each hydroxylase, UM-SCC-12, UM-SCC-11A, and NRK-52E cells were cultured to confluence. The cells were harvested after medium change. Transfer membranes of the cell lysates were probed with a CYP24A1 polyclonal antibody (Thermo Fisher Scientific, RRID: AB_11154312, CAT #PA521704), or a CYP27B1 polyclonal antibody (Abcam, Cambridge, UK, CAT #ab206655, Clone #EPR20271) and a monoclonal GAPDH antibody. Proteins were normalized to GAPDH. 

### 2.4. Local Production of Vitamin D_3_ Metabolites

#### 2.4.1. Vitamin D Metabolites

25-hydroxyvitamin D3-deuterated 3 used for internal standards was ordered from Cambridge Isotope Laboratories (Tewksbury, MA, USA). 1,25-dihydroxyvitamin D3-d3 was used as another internal standard (Sigma Aldrich). 24R,25-dihydroxyvitamin D3-d6, 25-hydroxyvitamin D3 monohydrate-d6, and 1,25-dihydroxyvitamin D3-d6 were used for the calibration standard curves (Sigma Aldrich). Liquid chromatography–mass spectroscopy (LCMS)-grade methanol and water were used to make mobile phases and to extract and purify samples (Thermo Fisher Scientific, Waltham, WA). Ammonium acetate, formic acid, and molecular grade ethanol were used to make mobile phases (Sigma Aldrich). 

#### 2.4.2. Cell Treatment

NRK-52E, UM-SCC-12, and UM-SCC-11A cells were cultured to confluence in 6 T75 flasks seeded at 10,000 cells/cm^2^. At confluence, the cells were treated with 5 × 10^−7^ M (0.203 μg/mL) 25(OH)D_3_-d6 for 24 h. The conditioned media were collected and extracted for HPLC to analyze the local secretion of vitamin D_3_ metabolites. Cell layers were lysed with Triton X-100 in 1x PBS to calculate DNA content. 

#### 2.4.3. Analytical Methods 

The production of 1,25(OH)_2_D_3_ and 24,25(OH)_2_D_3_ was measured using high-performance liquid chromatography–mass spectrometry as previously described [31]. Briefly, a 9-point standard curve was created from a range of 0.2-250 ng/mL from stock solutions of 24R,25-dihydroxyvitamin D3-d6, 25-hydroxyvitamin D3 monohydrate-d6, and 1,25-dihydroxyvitamin D3-d6. 1,25-dihydroxyvitamin D3-d3 and 25-hydroxyvitamin D_3_-d3 were used as internal standards. Calibration curves are shown in Appendix A.

The equations used to calculate vitamin D_3_ metabolites are in the supplementary methods. Measurements of 25(OH)D_3_, 1,25(OH)_2_D_3_, and 24,25(OH)_2_D_3_ in the conditioned media were compared to the 9-point standard curve equation, described below. The percent conversion of 25(OH)D_3_ to 1,25(OH)_2_D_3_ or 24,25(OH)_2_D_3_ was calculated as follows: % conversion = [(ng/mL metabolite)/(total 25(OH)D_3_ + 1,25(OH)_2_D_3_ + 24,25(OH)_2_D_3_)] × 100. The molarity of 1,25(OH)_2_D_3_ or 24,25(OH)_2_D_3_ in the conditioned media was calculated as follows: Molarity = grams/(volume × molecular weight).

Standard curves were generated as described below. Mass-to-charge ratios (423.55 > 121.18, 437.76 > 138.01, 407.47 > 107.21, 440.75 > 135.01) were used for quantification with cone and collision voltages between 13 and 17V and 14–18 V, respectively [26]. The calibration curve was calculated through weighted linear regression. The normalized peak area ratio of analyte to internal standard was plotted against the concentration of the 3 metabolites, 25(OH)D_3_, 1,25(OH)_2_D_3_, and 24,25(OH)_2_D_3_ [31]. The transitions were monitored for 62 ms.

### 2.5. Statistical Analysis

All data graphed are shown as means ± SEM. Treatment vs. control was assessed by means of the Wilcoxon matched-pairs signed rank test, with *p*-values ≤ 0.05 determined as significant. Baseline enzyme levels and vitamin D_3_ metabolite levels were determined significantly different between groups using one-way ANOVA with a Tukey post-test correction, with *p*-values ≤ 0.05 considered significant. Timepoint analyses were evaluated using a two-way ANOVA and a Tukey post-test, and *p*-values ≤ 0.05 were considered significant.

## 3. Results

### 3.1. Characterization of Estrogen Receptors

The Western blot analysis of laryngeal cancer cell lines UM-SCC-12 and UM-SCC-11A showed that these cell types differentially produce ERα66. UM-SCC-12 produced ERα66, ERα46, and ERα36 while UM-SCC-11A cells only produced ERα46 and ERα36 (Figure 1A,B, and Appendix A). Gene expression analysis using the primer targeting ERα66 but not its splice variants indicated the expression of ERα66 in only the UM-SCC-12 cells. The primer that targeted the seventh and eighth exon of ESR1, which is present in ERα66 and ERα46, demonstrated similar expression levels in both cell types. Similarly, the primer that targeted the ninth exon of ESR1, only found in ERα36, also measured similar expression levels in both cell types (Figure 1C). UM-SCC-11A cells had a greater expression of ERβ and GPR30 compared to UM-SCC-12 cells (Figure 1D,E). There was no difference in expression of the classical nuclear vitamin D receptor, VDR (Figure 1F).

### 3.2. Effect of 24R,25(OH)_2_D_3_ on Markers of Tumorigenesis In Vitro

UM-SCC-12 cells treated with 24R,25(OH)_2_D_3_ increased proliferation in a dose-dependent manner, with DNA synthesis at 10^−8^ M and 10^−7^ M concentrations being significantly higher than control (Figure 2A). A treatment/control analysis of three different experiments found increased cell proliferation in all doses examined (Appendix A). MTT activity was also increased at the highest concentration of 24R,25(OH)_2_D_3_ (Figure 2B). In contrast, UM-SCC-11A cells treated with 24R,25(OH)_2_D_3_ exhibited reduced proliferation in a dose-dependent manner, which was significant at 10^−7^ M (Figure 2C and Appendix A). MTT activity was similarly decreased (Figure 2D). 

In addition, 24R,25(OH)_2_D_3_ inhibited TUNEL staining in UM-SCC-12 cells, which was significant at 10^−8^ M and 10^−7^ M (Figure 2E and Appendix A). The two highest concentrations of 24R,25(OH)_2_D_3_ reduced the ratio of BAX/BCL2 mRNA expression (Figure 2F). P53 protein levels were also decreased in UM-SCC-12 cells treated with higher concentrations of 24R,25(OH)_2_D_3_ compared to the vehicle control (Figure 2G and Appendix A). Caspase 9 activity was reduced in a dose-dependent manner; the highest dose of 24R,25(OH)_2_D_3_ significantly inhibited its activity (Figure 2H). In contrast, in UM-SCC-11A cells, 24R,25(OH)_2_D_3_ increased TUNEL staining in a dose-dependent manner (Figure 2I and Appendix A). BAX/BCL2 mRNA expression increased only at the 10^−8^ M concentration compared to the vehicle and the 10^−9^ M concentration (Figure 2J). The highest concentration of 24R,25(OH)_2_D_3_ increased p53 levels in UM-SCC-11A cells compared to the vehicle and the 10^−9^ M concentration (Figure 2K and Appendix A). There was no change in caspase 9 activity in the UM-SCC-11A cells after treatment with 24R,25(OH)_2_D_3_ (Figure 2L).

24R,25(OH)_2_D_3_ stimulated metastasis in UM-SCC-12 cells. Nidogen-1 levels increased compared to control from treatment with the highest concentration of 24R,25(OH)_2_D_3_ (Figure 3A). In contrast, there was a decrease in nidogen-1 levels in the UM-SCC-11A cells (Figure 3B). The ratio of OPG/RANKL was reduced in UM-SCC-12 cells only at the 10^−8^ M concentration of 24R,25(OH)_2_D_3_ (Figure 3C), which indicates an increase in osteoclast activity. In contrast, the ratio was increased only in the highest concentration of 24R,25(OH)_2_D_3_ in UM-SCC-11A cells (Figure 3D). 

### 3.3. Characterization of Vitamin D_3_ Hydroxylase Enzymes

NRK-52E, UM-SCC-12, and UM-SCC22A cells all expressed mRNA for both hydroxylases. NRK-52E cells expressed the lowest level of CYP24A1 while UM-SCC-12 cells expressed the highest level. Although UM-SCC-11A cells expressed lower levels of CYP24A1 compared to UM-SCC-12 cells, the level of CYP24A1 was still higher than that of NRK-52E cells (Figure 4A). In contrast, the Western blot analysis showed that NRK-52E cells produced significantly higher amounts of CYP24A1 protein compared to UM-SCC-11A cells. There was no difference in the protein levels when comparing the levels from either cell type to UM-SCC-12 cells (Figure 4B). Both UM-SCC-12 and UM-SCC-11A cells expressed higher levels of CYP27B1 compared to NRK-52E cells (Figure 4C). The Western blot analysis showed higher protein levels of CYP27B1 in UM-SCC-11A compared to UM-SCC-12 and NRK-52E cells (Figure 4D). 

### 3.4. Production of Vitamin D_3_ Metabolites

The elution times for vitamin D_3_ metabolites in our system are presented in Appendix A. The peak analysis of all measured metabolites in cultures of both UM-SCC-12 (Figure 5A) and UM-SCC-11A (Figure 5B) cells shows a clear separation between the vitamin D_3_ metabolites. After treatment with 25(OH)D_3_ for 24 h, conditioned media were assessed by means of HPLC for 1,25(OH)_2_D_3_ and 24,25(OH)_2_D_3_ production. NRK-52E cells produced significantly less 24,25(OH)_2_D_3_ than UM-SCC-12 cells and UM-SCC-11A cells. UMSCC-11A cells produced the most 24,25(OH)_2_D_3_ at approximately 23 ng/mL 24,25(OH)_2_D_3_, while UM-SCC-12 produced about 15 ng/mL (Figure 6A). This led to a 1%, 14.6%, and 26% conversion of 25(OH)D_3_ to 24,25(OH)_2_D_3_ in NRK-52E, UM-SCC-12, and UM-SCC-11A cells, respectively (Figure 6B). While NRK-52E cells produced about 10^−9^ M 24,25(OH)_2_D_3_, UM-SCC-12 and UM-SCC-11A cells produced significantly more with 3.5 × 10^−8^ M and 5.6 × 10^−8^ M, respectively (Figure 6C). All three cell types produced similar levels of 1,25(OH)_2_D_3_, with approximately 1ng/mL measured in the media (Figure 6D). About 1% of the treated 25(OH)D_3_ was converted into 1,25(OH)_2_D_3_, equating to about 2 × 10^−9^ M 1,25(OH)_2_D_3_ (Figure 6E,F).

## 4. Discussion

Vitamin D_3_ supplementation has attracted much attention in cancer therapeutics, although its potential in laryngeal cancer has not been investigated. The contradictory reports of laryngeal cancer as a hormone-responsive cancer and vitamin D_3_ deficiency as a risk factor make distinguishing the potential for vitamin D_3_ treatment challenging. This study and previous studies show that many factors are at play to account for these contradictory findings. In female breast cancer, the presence or absence of ERα66 changes the cell’s response to treatment with vitamin D_3_ metabolites [33]. Our lab showed that some male laryngeal cancers are responsive to estrogen, opening the possibility of this factor playing a significant role in the response of male cells to metabolites of vitamin D_3_ [16]. Classifying laryngeal cancer as estrogen-responsive makes using ERα66 as an early diagnosis marker and a prognostic marker plausible. The high reactivity of estrogen receptor alpha was found only in malignant laryngeal tissue and not the surrounding tissue [6,13]. These findings make ERα66 a promising candidate as a marker for malignancy in the future.

ERα66 has long been used as a critical biomarker for determining aggression and the course of treatment in breast cancer. The prognostic value of ERα in male laryngeal cancer is becoming more apparent. More recent studies demonstrate the role of ERα36 in tumorigenesis by regulating metastasis and invasion, as well as resistance to hormone therapies like tamoxifen. Finally, the role of ERα46 in tumorigenesis needs further investigation.

We hypothesize that estrogen-responsive cancers will behave similarly and that the expression of ERα66 will change the response to vitamin D_3_ treatment. Previous studies in our lab have already demonstrated the potential of ERα66 as an indicator of aggression in male laryngeal cancer, similar to female breast cancer [16,25]. Our Western blot analysis showed that different laryngeal squamous-cell carcinoma cells produce different levels of ERα66. To evaluate the role of ERα66 in mediating the effects of 24R,25(OH)_2_D_3_, we used two different male laryngeal cancer cell lines that produce similar levels of ERα36 and ERα46, but one cell type, UM-SCC-12 cells, also has ERα66. The differential expression of ERα66 emphasizes its role in mediating the effects observed because there is no change in the levels of ERα36 or ERα46. We observed a difference in expression of ERβ and GPR30, so the role of these receptors in mediating the effects cannot be ruled out. The similar expression of VDR observed in both cell types makes it unlikely that VDR plays a role in mediating the effect of 24R,25(OH)_2_D_3_. However, as we have observed in breast cancer, cells that differentially express ERα66 respond differently to the vitamin D_3_ metabolite, 24R,25(OH)_2_D_3_ [31], making its role in mediating laryngeal cancer progression more plausible. Therefore, we first wanted to evaluate the cells’ response to 24R,25(OH)_2_D_3_.

We examined the therapeutic potential of 24R,25(OH)_2_D_3_ in laryngeal cancer cells by treating UM-SCC-12 and UM-SCC-11A cells with vehicle or 24R,25(OH)_2_D_3_. 24R,25(OH)_2_D_3_ increased markers of tumorigenesis in the ERα66-positive UM-SCC-12 cells by increasing indicators of proliferation and metastasis and reducing apoptosis. In contrast, 24R,25(OH)_2_D_3_ reduced markers of tumorigenesis in the ERα66-negative UM-SCC-11A through a reduction in markers of proliferation and metastasis and an increase in apoptosis. Overall, because these cells differentially express ERα66, these data demonstrate that the effect of 24R,25(OH)_2_D_3_ on laryngeal cancer is related to the expression of ERα66. However, we also observed a difference in the expression of ERβ and GPR30, which may also play a role. Interestingly, in ERα66-positive MCF7 breast cancer cells treated with 24R,25(OH)_2_D_3_, although proliferation was stimulated, the increase in apoptosis and the reduction in various metastatic markers demonstrated an overall antitumorigenic effect, while in ERα66-negative HCC38 breast cancer, 24R,25(OH)_2_D_3_ increased tumorigenesis [33]. The similar expression levels of ERβ and GPR30 in both breast cancer cells further suggest that ERa66 is the major player regulating the response to 24R,25(OH)_2_D_3_, which could be responsible for the difference in both cancers. The opposite effects observed in laryngeal cancer versus breast cancer may be due to the difference in male cells compared to female cells. 

These data suggest that the expression of ERα66 plays a critical role in the cellular response to the vitamin D_3_ metabolite. However, while there was a difference in cellular response to 24R,25(OH)_2_D_3_ based on ERα66 expression, it was unclear as to why these responses were different in these two different cancer types even when expressing similar levels of ERα66. This could reflect a fundamental difference between breast cancer and laryngeal cancer. Another possible reason for the difference could be sex-dependent; the breast cancer cells originated from female donors and the laryngeal cancer cells were taken from male donors. Alternatively, factors other than ERα66 may contribute to the local regulation of 24R,25(OH)_2_D_3_ through autocrine signaling to mediate its effect on tumorigenesis. For example, ERβ and GPR30 were present in the cells and may interact with ERα isoforms; similarly, the androgen receptor may play a role.

Based on previous data, demonstrating the regulation of vitamin D3 hydroxylases by estrogen [42], as well as the ability of laryngeal cancer cells to produce their own estrogen [25], suggested the potential for an autocrine-mediated response to 24R,25(OH)_2_D_3_. Therefore, we aimed to determine the if the laryngeal cancer cells can produce vitamin D_3_ metabolites locally, which could induce an autocrine effect on the cells. Our results confirm that laryngeal cancer cells can convert 25(OH)D_3_ into active metabolites. Both cell types expressed the 24-hydroxylase, CYP24A1, and the 1−hydroxylase, CYP27B1, demonstrating the potential for producing active vitamin D_3_ metabolites. CYP24A1 is not only responsible for the hydroxylation of 25(OH)D_3_ to 24,25(OH)_2_D_3_ but also the hydroxylation of 1,25(OH)_2_D_3_ into 1,24,25(OH)_3_D_3_. A high expression of CYP24A1 is known as the initial step in the degradation of the 1,25(OH)_2_D_3_ pathway [31,35,43]. The ERα66-positive UM-SCC-12 cells expressed significantly higher levels than the ERα66-negative UM-SCC-11A cells, and both cell types expressed more elevated levels than the positive control NRK-52E cells. However, when evaluating the cells for protein production, NRK-52E cells produced the most CYP24A1, significantly more than UM-SCC-11A cells. CYP27B1 expression showed higher levels in both laryngeal cancer cells than NRK-52E cells. However, the expression did not change with ERα66 expression. In contrast, the protein levels of CYP27B1 increased in ERα66-negative UM-SCC-11A cells compared to UM-SCC-12 cells. Together, these data suggest that laryngeal cancer’s ability to produce vitamin D_3_ metabolites in response to 25(OH)D_3_ may depend on the expression of ERα66. The local production of active vitamin D_3_ metabolites can affect the cells through autocrine/paracrine signaling [34,35]. It is also interesting to note that the cells examined show similar levels of 1,25(OH)_2_D_3_ production, while the production of 24R,25(OH)_2_D_3_ was cell type-dependent, which may indicate the vital role of 24R,25(OH)_2_D_3_ on cell tumorigenesis and as an endocrine factor.

Laryngeal cancer cells and breast cancer cells produced 24,25(OH)_2_D_3_ and 1,25(OH)_2_D_3_ at levels that were greater than are present in serum [31,44,45]. Moreover, the locally produced metabolites were at levels corresponding to the concentrations at which we noted cellular responses [45]. Both laryngeal cancer cell lines produced significantly more 24,25(OH)_2_D_3_ than NRK-52E cells in response to 25(OH)D_3_, and UM-SCC-11A cells produced substantially more 24,25(OH)_2_D_3_ compared to UM-SCC-12 cells. All three cell lines produced similar levels of 1,25(OH)_2_D_3_, significantly higher than the 30 pmol/L serum levels measured in patients. UM-SCC-11A cells produced the highest amount of 24,25(OH)_2_D_3_, similar to MCF7 cells [31]. Interestingly, while MCF7 cells are ERα66-positive, they responded similarly to 24R,25(OH)_2_D_3_ as UM-SCC-11A cells in that tumorigenesis was reduced [33]. UM-SCC-12 cells, while ERα66-positive, responded similarly to the ERα66-negative breast cancer cell line HCC38. 

It is important to note that the expression and protein levels of the CYP24A1 and CYP27B1 enzymes are not directly correlated with the production of the local metabolites. The expression and protein levels do not measure enzyme activity and, therefore, cannot be compared to the production of the vitamin D_3_ metabolites. The presence of the enzymes is used to indicate the cells’ ability to produce metabolites. These findings suggest that the response to 24R,25(OH)_2_D_3_ may be dependent on the local production of the metabolite which is related to the presence of ERα66 and the sex of the cells.

In summary, 24R,25(OH)_2_D_3_ can regulate markers of laryngeal cancer tumorigenesis and its effect is dependent on the expression of ERα66, similar to that observed in breast cancer. These findings demonstrate a potential therapeutic agent for aggressive laryngeal cancer cells that do pose a risk to calcium levels, making it more plausible as an anti-tumor agent. Through the expression of CYP24A1 and CYP27B1, laryngeal cancer cells can produce vitamin D_3_ metabolites 1,25(OH)_2_D_3_ and 24,25(OH)_2_D_3_ at biologically relevant levels. The local production of these metabolites suggests the potential autocrine or paracrine effects of these metabolites on tumorigenesis. 24R,25(OH)_2_D_3_ may be used as a potential therapeutic agent in some laryngeal cancers; however, the local production and the level of ERα66 must be considered. This study did not measure the production of other vitamin D_3_ metabolites, so their role in laryngeal cancer tumorigenesis cannot be ruled out. The similar expression of ERα36 and ERα46 in the two cell lines eliminates the possibility of these receptors playing a role in this effect, but the role of the other active receptor, ERβ, cannot be dismissed. Further investigation is needed to address these possibilities. To date, the use of vitamin D_3_ supplementation in laryngeal cancer has not yet been examined.

## 5. Conclusions

In conclusion, the strong association of estrogen receptors, specifically ERα66, with malignancy observed in laryngeal tissue, along with our data demonstrating the differential responses of ERα66-positive and ERα66-negative cells to treatments like the vitamin D_3_ metabolite 24R,25(OH)_2_D_3_, emphasize the potential of ERα isoform levels as an early-stage prognostic and diagnostic biomarker, as well as for planning treatment modalities. While evaluating the estrogen receptor status will be beneficial in predicting the tumor’s response to vitamin D_3_ supplementation, there is still a possibility of a sex-dependent factor. Further studies will need to be performed to discern how sex hormones can play a role in a cell’s ability to produce vitamin D_3_ metabolites by comparing breast cancer and laryngeal cancer cells and including female laryngeal cancer cells. These data also emphasize the role of 24R,25(OH)_2_D_3_ as a therapeutic agent in laryngeal cancer tumorigenesis as a less aggressive treatment modality than the current standard chemotherapy drugs. Our data indicate that laryngeal cancer cells can locally produce vitamin D_3_ metabolites, which can regulate tumor development, an effect that is related to the level of ERα66. This is the first study demonstrating the therapeutic potential of vitamin D_3_ metabolites in laryngeal cancer, the importance of using ERα66, and the levels of vitamin D_3_ metabolites in the localized tissue as predictive markers.

## Figures and Tables

**Figure 1 cancers-16-01635-f001:**
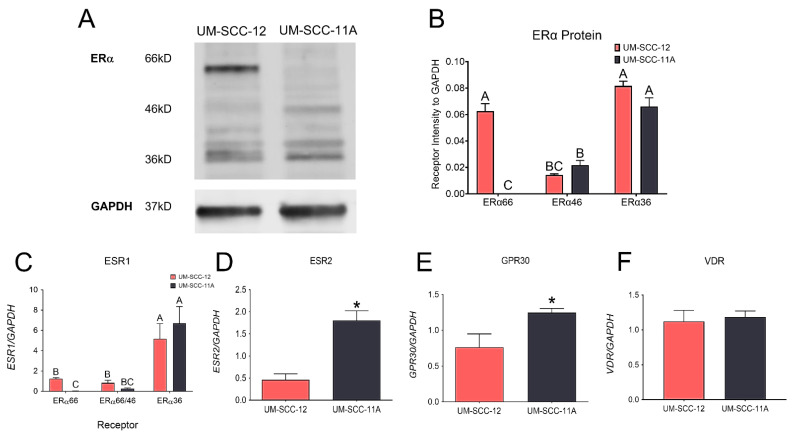
Analysis of estrogen receptor levels. Western blot analysis measured baseline protein levels of ERα66, ERα46, and ERα36 in UM-SCC-12 and UM-SCC-11A cells (n = 3/cell type) (**A**). Quantification of Western blot analysis (**B**). Gene expression of ERα66, ERα46/66, and ERα36 in UM-SCC12 and UM-SCC-11A cells (**C**). Quantification of estrogen receptor alpha levels is presented as the mean ± standard error of n = 6 per group. Groups not sharing a letter are statistically different at *p* < 0.05 by means of two-way ANOVA with a Tukey post-test. Gene expression levels of ERβ (**D**), GPR30 (**E**), and the vitamin D receptor (**F**) in UM-SCC-12 and UM-SCC-11A cells. Groups labeled with an asterisk (*) are statistically significant compared to UM-SCC-12 expression by means of Student’s *t*-test with a Tukey post-test. Graphs are presented as the mean ± standard error of n = 6 per group. All graphs are from one of two independent experiments, both with comparable results.

**Figure 2 cancers-16-01635-f002:**
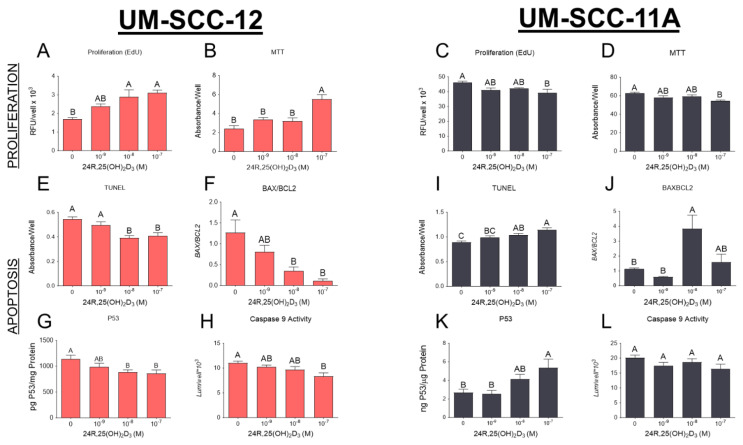
Analysis of 24R,25(OH)_2_D_3_ effect on markers of tumor growth and apoptosis. UM-SCC-12 and UM-SCC-11A cells were treated with vehicle, 10^−9^ M, 10^−8^ M, or 10^−7^ M 24R,25(OH)_2_D_3_ for 15 min, and then media were replaced with fresh media without the vitamin D metabolite for 20 h to perform the EdU assay in UM-SCC-12 cells (**A**). The MTT assay was performed 24 h after the treatment of UM-SCC-12 cells (**B**). UM-SCC-11A cells were assessed for EdU incorporation 20 h after treatment (**C**). The MTT assay was performed 24 h after UM-SCC-11A cells were treated with 24R,25(OH)_2_D_3_ (**D**). A TUNEL assay was used to measure DNA fragmentation 24 h after treatment in UM-SCC-12 cells (**E**). The gene expression of the ratio between BAX and BCL2 was assessed in UM-SCC-12 cells 12 h after treatment (**F**). p53 protein levels were measured using ELISA in UM-SCC-12 cells (**G**) 24 h after treatment. Caspase 9 activity was measured in UM-SCC-12 cells 24 h after treatment (**H**). Similar measurements were made in UM-SCC-11A cells for TUNEL (**I**), BAX/BCL2 mRNA ratio (**J**), p53 levels (**K**), and Caspase 9 activity (**L**). Data are presented as the mean ± standard error of n = 6 independent cultures per group. Groups not sharing a letter are statistically different at *p* < 0.05 by means of one−way ANOVA with a Tukey post-test.

**Figure 3 cancers-16-01635-f003:**
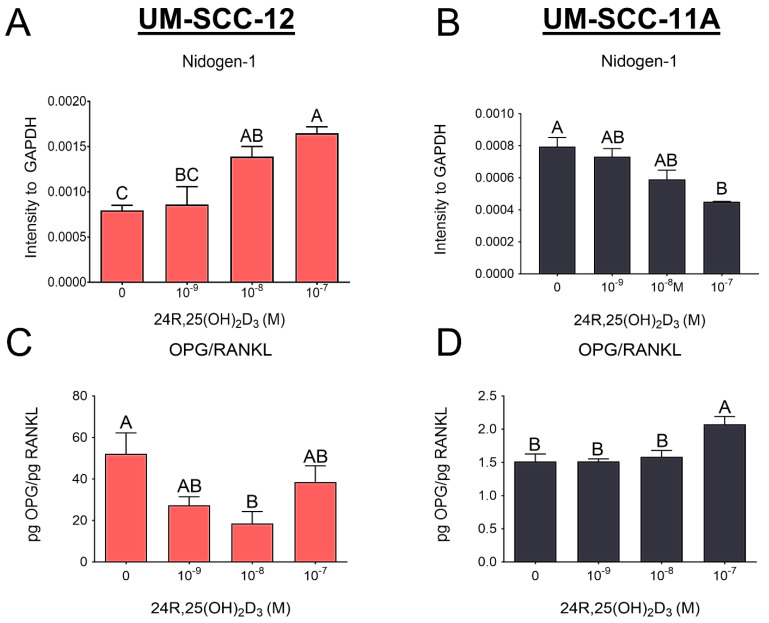
Analysis of 24R,25(OH)_2_D_3_ effect on markers of metastasis. The Western blot analysis of lung metastasis marker, Nidogen-1, was performed 24 h after treatment in UM-SCC-12 cells (**A**) and UM-SCC-11A cells (**B**). The effect of 24R,25(OH)_2_D_3_ on the osteoclast activity indicator OPG/RANKL ratio was measured using ELISAs 24 h after treatment in UM-SCC-12 cells (**C**) and UM-SCC-11A cells (**D**). Groups not sharing a letter are statistically different at *p* < 0.05 by means of one-way ANOVA with a Tukey post-test. Data are presented as the mean ± standard error of n = 6 independent cultures per group. Data are from a single experiment; all experiments were repeated to ensure the validity of the results.

**Figure 4 cancers-16-01635-f004:**
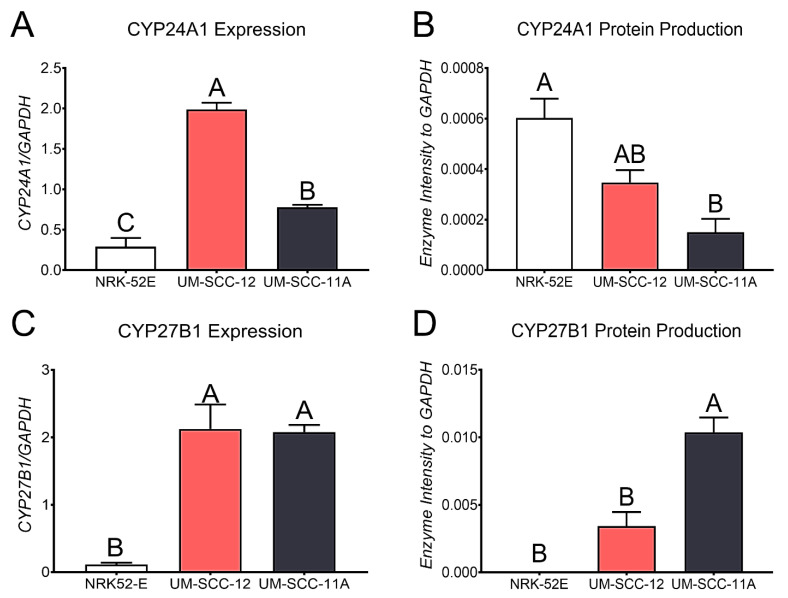
Baseline analysis of enzymes 1α-hydroxylase and 24-hydroxylase. Baseline mRNA expression levels of 24-hydroxylase/CYP24A1 (**A**) measured 12 h after final feedings, and protein levels measured by means of Western blot in NRK-52E, UM-SCC-12, and UM-SCC-11A cells 24 h after the final feeding (**B**). Gene expression of 1a-hydroxylase/CYP27B1 measured using RT qPCR (**C**) and protein levels measured by means of Western blot (**D**). Gene expression data are presented as the mean ± standard error of n = 6 samples per group while the Western blot analysis has n = 3 per group. Groups not sharing a letter are statistically different at *p* < 0.05 by means of one-way ANOVA with a Tukey post-test.

**Figure 5 cancers-16-01635-f005:**
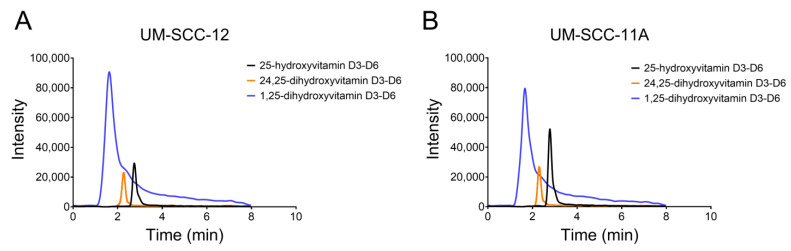
HPLC-MS analysis of 24,25(OH)2D3 production. Peak analysis of 25-hydroxyvitamin d3–d6, 1,25-dihydroxyvitamin d3–d6, and 24,25-dihydroxyvitamin d3–d6 observed in UM-SCC-12 cells (**A**) and UM-SCC-11A cells (**B**).

**Figure 6 cancers-16-01635-f006:**
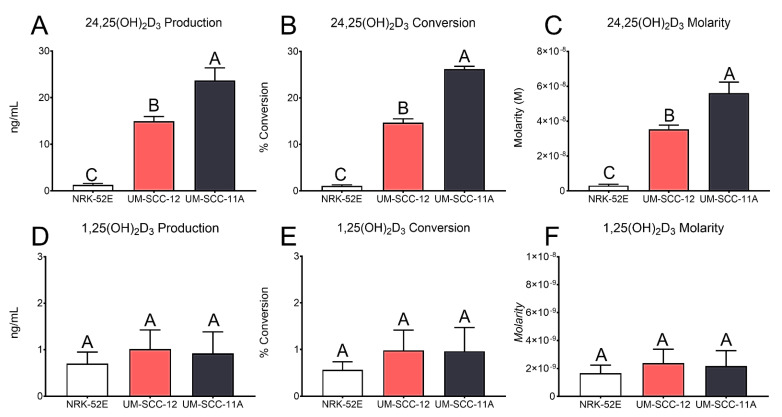
HPLC-MS analysis of 24,25(OH)_2_D_3_ and 1,25(OH)_2_D_3_ production. Cells were treated with 0.0203 μg/mL 25(OH)D_3_−d6 in 10 mL and assessed for 24,25(OH)_2_D_3_ and 1,25(OH)_2_D_3_ production. The data are shown as the total amount of 24,25(OH)_2_D_3_ per mL of 25(OH)D_3_ used as treatment (**A**), the percentage of 25(OH)D_3_ converted to 24,25(OH)_2_D_3_ (**B**), and the molar concentration of 24,25(OH)_2_D_3_ measured (**C**). The total amount of 1,25(OH)_2_D_3_ per mL of 25(OH)D_3_ used as treatment (**D**), the percentage of 25(OH)D_3_ converted to 1,25(OH)_2_D_3_ (**E**), and the molar concentration of 1,25(OH)_2_D_3_ measured (**F**). Data shown are from a single representative experiment of two repeats and are presented as the mean ± standard error of six independent cultures per treatment group. Groups not sharing a letter are statistically different at *p* < 0.05 by means of one-way ANOVA.

## Data Availability

The data generated in this study are available upon reasonable request from the corresponding author.

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
