# Peer review of "Laryngeal Cancer Cells Metabolize 25-Hydroxyvitamin D3 and Respond to 24R,25-dihydroxyvitamin D3 via a Mechanism Dependent on Estrogen Receptor Levels"

_cancers, 2024, doi:10.3390/cancers16091635_

Round 1
Reviewer 1 Report
Comments and Suggestions for Authors
The study is well organized and starts from the information that claims that vitamin D3 deficiency is associated with an increased risk of malignancy in cancer. Through the experimental model approach, the authors try to define a relationship between the expression of estrogen receptors and the production of vitamin D3 metabolites in laryngeal cancer cells. By using an appropriate methodology, was evaluated the effect of 24R,25(OH)2D3 treatment on laryngeal tumorigenesis markers and estrogen receptor expression as well as cellular functionality in two different laryngeal cancer cell lines, UM-SCC-12 ER66 cells -positive and ER66-negative UM-SCC-11A cells. In addition, to assess whether vitamin D metabolites could act through autocrine/paracrine mechanisms, CYP24A1 and CYP27B1 protein expression were evaluated. The results show that treatment with 24R,25(OH)2D3 increased tumorigenesis markers in ER66-positive UM-SCC-12 cells by increasing proliferation and metastasis indicators associated with reducing apoptosis. The data presented demonstrate the therapeutic potential of vitamin D3 metabolites in laryngeal cancer and the importance of using ER66 and vitamin D3 metabolite levels as predictive markers.
The manuscript is interesting due to its original approach to the topic. The presented data adequately support the authors' conclusions.
However, in the final part of the article, the authors should more clearly highlight the most relevant conclusions of the study and indicate future directions.
Overall, the article could be a substantial contribution to the journal. Therefore, I recommend the manuscript for publication after minor changes and updates by the authors have been considered.

Author Response
Response to Reviewer 1
- The study is well organized and starts from the information that claims that vitamin D3 deficiency is associated with an increased risk of malignancy in cancer. Through the experimental model approach, the authors try to define a relationship between the expression of estrogen receptors and the production of vitamin D3 metabolites in laryngeal cancer cells. By using an appropriate methodology, was evaluated the effect of 24R,25(OH)2D3 treatment on laryngeal tumorigenesis markers and estrogen receptor expression as well as cellular functionality in two different laryngeal cancer cell lines, UM-SCC-12 ER66 cells -positive and ER66-negative UM-SCC-11A cells. In addition, to assess whether vitamin D metabolites could act through autocrine/paracrine mechanisms, CYP24A1 and CYP27B1 protein expression were evaluated. The results show that treatment with 24R,25(OH)2D3 increased tumorigenesis markers in ER66-positive UM-SCC-12 cells by increasing proliferation and metastasis indicators associated with reducing apoptosis. The data presented demonstrate the therapeutic potential of vitamin D3 metabolites in laryngeal cancer and the importance of using ER66 and vitamin D3 metabolite levels as predictive markers. The manuscript is interesting due to its original approach to the topic. The presented data adequately support the authors' conclusions.
We thank the reviewer for their kind review of our paper.
- However, in the final part of the article, the authors should more clearly highlight the most relevant conclusions of the study and indicate future directions.
A conclusion section was added to the end of the manuscript to clearly highlight our conclusions and include future directions.
Reviewer 2 Report
Comments and Suggestions for Authors
The topic of the reviewed study deals with known disproportion of male/female susceptibility to laryngeal cancer. First it was explained by higher men inclination to tobacco smoking and alcohol abusing (primary carcinogenic factor in laryngeal cancer). However when the groups with same addiction rate were compared disproportion remained. Then an attention was turned (among other suppositions) on sex hormones. The authors have done a large study focused on estrogen receptors in association with vitamin D3.
In principle I accept the experimental protocol as well as final conclusions. Nevertheless I am having a few remarks to be answered/commented before acceptance.
1. Several times you referred to literature results coming from breast cancer. Why breast cancer cell line was not included into experiments? Then comparison would be much stronger than literature data.
2. There are two sections titled “cell culture”. UM-SSC12 and UM-11A are described first but animal cell line NRK-52E appeared later without information about origin.
3. SCC ell lines were purchased from Carey Lab… Is it associated with Tom Carey, once (still?) a giant person in head and neck cancer biology?
4. The findings of the study are dedicated to diagnosis of early laryngeal cancer (line 97-98, 476). Still a vast majority of laryngeal cancers subject are diagnosed in advanced stage. So, the finding goes to vain. On the other hand an association of ER with aggressiveness seems to be well proven.
5. References list requires another look of control quality eye. Look at positions: 11 (volume, pages lacking), 15 (pages missing), 17 (volume, pages missing), 25, 29, 35 (pages missing).
6. I think this is rather editorial office job to remove some typing errors (line 29, full stop missing, line 120 (extra space), line 500 (fragment /?/ missing).
Minor revision suggested.
Author Response
Response to Reviewer 2
- Several times you referred to literature results coming from breast cancer. Why breast cancer cell line was not included into experiments? Then comparison would be much stronger than literature data.
The focus of this paper emphasized the comparison between ERa66 positive and ERa66 negative laryngeal cancer cells. Including breast cancer cell lines would introduce further variables not examined in this study. Since the data of cancer cell line was published, we edited the discussion to emphasis the comparisons between the two kinds of tumors.
- There are two sections titled “cell culture”. UM-SSC12 and UM11A are described first but animal cell line NRK-52E appeared later without information about origin.
Information regarding NRK-52E cells was added to the cell culture section. The second section was relabeled “Cell Treatment” distinguishing between normal culture conditions and treatment protocols.
- SCC ell lines were purchased from Carey Lab… Is it associated with Tom Carey, once (still?) a giant person in head and neck cancer biology?
Yes, these cell lines were purchased directly from Dr. Thomas Carey. We are honored to have been able to work with such a remarkable researcher in the head and neck cancer biology field.
- The findings of the study are dedicated to diagnosis of early laryngeal cancer (line 97-98, 476). Still a vast majority of laryngeal cancers subject are diagnosed in advanced stage. So, the finding goes to vain. On the other hand an association of ER with aggressiveness seems to be well proven.
Functional estrogen receptors were observed in laryngeal cancer tissue with a high reactivity both at early and advance stage compared to healthy tissue. The use of estrogen receptor alpha expression, specifically estrogen receptor alpha 66 may not only be used as a biomarker for aggression but also for early-stage detection. However, the existence of the receptor can be used also for the decision on treatment modality. This point was emphasized in the conclusion.
- References list requires another look of control quality eye. Look at positions: 11 (volume, pages lacking), 15 (pages missing), 17 (volume, pages missing), 25, 29, 35 (pages missing).
References were edited and updated for consistency.
- I think this is rather editorial office job to remove some typing errors (line 29, full stop missing, line 120 (extra space), line 500 (fragment /?/ missing).
These errors were corrected.